# Psychosocial Screening in Adult Burns Inpatients within a Scottish Burns Unit

**Dawn Lindsay \*, Kim Kirkwood and Rebecca Crawford**

Burns Unit, NHS Greater Glasgow and Clyde, Glasgow Royal Infirmary, 84 Castle Street, Glasgow G4 0SF, UK; kim.kirkwood@ggc.scot.nhs.uk (K.K.); rebecca.crawford@ggc.scot.nhs.uk (R.C.)
\* Correspondence: dawn.lindsay4@ggc.scot.nhs.uk

**Abstract:** National Burns Care Standards (NBCS) within the UK recommend psychological care throughout the burn pathway and psychosocial screening of inpatients admitted for over 24 h, at a time when this is clinically appropriate and prior to discharge. This brief report presents preliminary data from an audit of psychosocial screening in adult burns inpatients within a Scottish Burns Unit over a three-year period. Results are reported on the frequency and type of psychosocial screening completed. Differences between the groups of inpatients who were screened and those not screened are presented and discussed with a focus on plans for increasing the number of inpatients screened and improvements in how psychosocial screening data is collected.

**Keywords:** psychological screening; psychosocial; burns; clinical psychology; inpatient care; mental health





## 1. Introduction

Burn injuries can have an enduring and detrimental effect on an individual's physical and psychological wellbeing and on quality of life [1]. Adults sustaining burns injuries are known to have higher rates of pre-existing mental health problems. These pre-existing difficulties are associated with poorer psychosocial adjustment in the aftermath of a burn injury [2,3]. This suggests a level of vulnerability within this population [3]. Even in those with no pre-existing mental health problems, experiencing distress whilst being hospitalised can lead to delayed rates of recovery [4]. Poor psychosocial adjustment and quality of life has been linked to a higher prevalence of depression, post-traumatic stress disorder (PTSD), anxiety, and challenges related to changes in physical appearance and loss of function amongst adults with burns injuries [1,5–7].

National Burns Care Standards within the UK (NBCS) [8] recommend routine psychological care and assessment throughout the burn pathway, with similar guidelines followed in Europe [9]. NBCS also recommend psychosocial screening of inpatients admitted for over 24 h, at a time when this is clinically appropriate and prior to discharge. Psychosocial screening in the adult burns patient population is recommended to allow psychological care to be preventative by identifying those with arising psychosocial difficulties and to help normalise psychological reactions after a burns injury [10]. Screening has been found to be feasible to implement and can identify distress amongst patients from the early stages of hospitalisation [10] to outpatient settings [11,12].

NBCS recommend that appropriate treatment should be given based on psychosocial needs identified through screening [1]. Three levels of psychological care have been identified: (1) basic screening and gauging psychological need; (2) psychological assessment, psychoeducation and low-level intervention such as emotional care; (3) psychological therapy and signposting to other appropriate services [10,12]. Utilising a tiered approach to assessing and supporting psychological needs throughout the pathway can embed clinical psychology within burns multidisciplinary teams (MDT) and highlight its value [12].

The use of a tiered approach within psychosocial screening with burns inpatients has been found to be cost-effective and to guide appropriate outcomes [13]. Methods of screening have been identified, which incorporate a range of psychosocial professionals including nursing staff, assistant psychologists, qualified clinical psychologists and psychiatry. Face-to-face psychosocial screening and indirect discussions at burns MDT meetings were found to be comparable in identifying those who may benefit from further psychological assessment [14]. Each method identified individuals with psychological needs that the other did not, suggesting that multiple methods of screening may be beneficial. It is suggested that screening tools be developed by services to assess the wide range of difficulties known to be prevalent amongst the burns population [10] and that brief measures may relieve the burden on patients and staff [15]. Research on methods of psychosocial screening and screening tools suggests that there is complexity and variation in implementing psychosocial screening across burns services.

This audit presents data gathered from the Burns Unit in Glasgow, which also hosts the newly commissioned National Burns Service for Scotland, providing care for all major adult and paediatric burns in Scotland. As part of evolving into the National Burns Service, the unit is focused on meeting NBCS standards, including psychosocial screening, assessment and treatment. The aims of this paper are to: (1) report on preliminary data exploring the current frequency and methods of psychosocial screening for adult burns inpatients; (2) investigate any differences between those inpatients who completed a psychosocial screening and those who were not screened.

## 2. Materials and Methods

### 2.1. Design

This brief report presents preliminary data from an audit of routine clinical data of burns inpatient admissions gathered over a three-year period. A database was developed by collating anonymised data from paper and electronic note systems, and the data was analysed retrospectively. The electronic clinical note system consists of notes written by professionals directly involved in an individual's care and scanned copies of paper notes and documents completed on the ward. Data recorded in the audit included demographics, burn-injury information, and the frequency and method of psychosocial screening. Data were gathered by reading electronic note entries, looking through scanned copies of weekly MDT forms and checking for any scanned screening questionnaires/measures. Many of the scanned documents had handwritten information. The study was classed as a clinical audit by the affiliated health board.

The National Burns Service in Scotland does not currently report data about psychosocial screening on the International Burn Injury Database (iBID) [16]. However, the coding system for gathering data about methods of screening was used as a framework for this audit. Direct consent was not obtained from patients, as the audit made use of anonymised and routinely collected clinical data.

### 2.2. Participants and Setting

The audit comprised 460 adult inpatients admitted to a Scottish National Burns Unit for specialist treatment and management between November 2019 and September 2022. The burns MDT includes medical and nursing staff, physiotherapists, occupational therapists and pharmacists. Within the burns MDT, the burns clinical psychology team holds responsibility for coordinating psychosocial screening. They are embedded within the MDT and attend weekly meetings, which includes discussions about the treatment plan for each inpatient. Psychosocial discussions are currently a standard part of these meetings. During the reported audit period, there was one clinical psychologist who had completed doctoral-level professional training, working part-time within the MDT (0.3 WTE). Psychosocial screening completed by the burns clinical psychologist involves a direct interview to assess the presence of psychological distress, including depression, anxiety, appearance concerns and trauma-related symptoms.

An on-site mental health liaison service provides screening for inpatients who present with suicidal or self-harming thoughts and behaviours and for those with pre-existing mental health problems. This team consists of psychiatry, psychology and nursing staff. There is access to a hospital addictions team who assess and treat inpatients requiring support for substance and alcohol dependence whilst in hospital. They do not directly provide screening and assessment for mental health and other psychosocial difficulties. As such, this is not included in the recording of the frequency and methods of psychosocial screening completed.

*2.3. Measures*

One of the aims of this audit was to gather preliminary data about the frequency and methods of psychosocial screening currently used. Nursing and medical staff had been encouraged by the clinical psychology team to administer the Patient Health Questionnaire-4 (PHQ-4) during the initial assessment when a patient was first admitted to the ward. The PHQ-4 is a 4-item questionnaire measuring the frequency of two depressive symptoms (PHQ-2) and two anxiety symptoms (Generalised Anxiety Disorder-2 (GAD-2)). It has been shown to be a valid and ultra-brief screening tool for depression and anxiety [17] and has been used in the adult burns patient population [10] as part of psychosocial screening.

*2.4. Data Analysis*

Descriptive statistics were used to record the number and method of inpatients who had been screened. Non-parametric statistical methods, Mann–Whitney U tests and Chi–Square tests were used to explore whether there was any significant difference between those who were screened and those who were not screened.

**3. Results**

*3.1. Psychosocial Screening*

Of the 460 inpatients recorded, 55 (12%) were admitted for less than 24 h and excluded as not being appropriate to screen. A further seven inpatients (2%) were excluded for not being clinically appropriate to screen (e.g., when a patient has died, is too physically unwell or has declined to be screened). This left 398 inpatients eligible for psychosocial screening. A total of 132 (33%) patients were coded as having some level of screening documented within clinical notes, and 266 patients (67%) had no psychosocial screening.

There were three methods of psychosocial screening completed: a PHQ-4 questionnaire administered by nursing staff, direct interview with the burns clinical psychologist and direct interview with the mental health liaison service. Additionally, there were instances when multiple methods of screening were completed. Figure 1 shows the combinations of methods of screening completed. Overall, 83 (63%) inpatients were screened by direct interview by the burns clinical psychologist and/or the mental health liaison service. Screenings completed by the mental health liaison service included 14 inpatients who had presented with burns sustained by deliberate self-harm.

*3.2. Demographics*

Data were further analysed to explore whether there were any significant differences between those screened and those not screened. Demographic information is shown in Table 1.

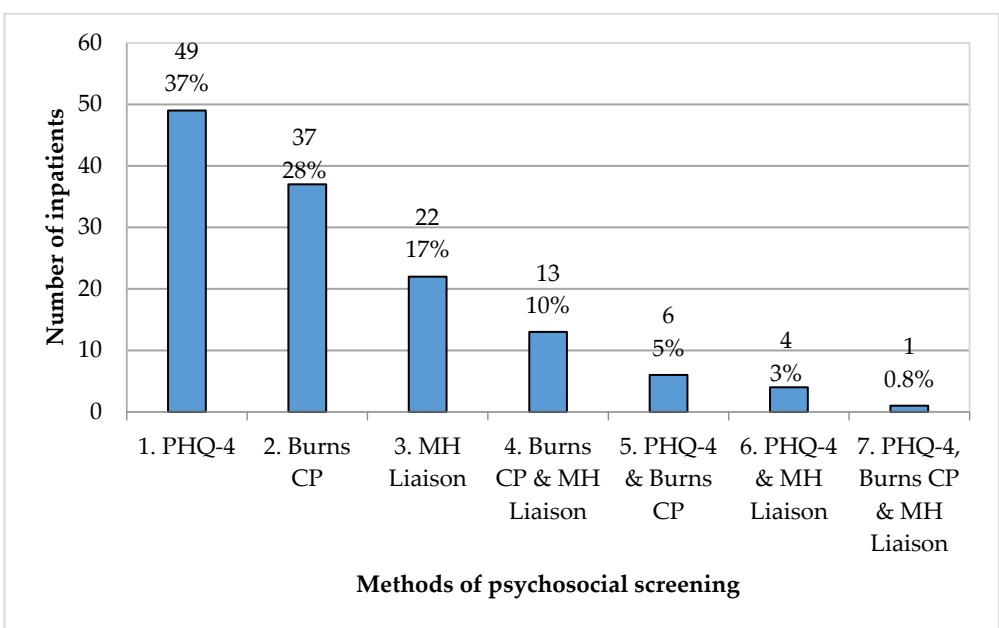

**Figure 1.** Methods of psychosocial screening. Codes for screening type: 1. PHQ-4 = PHQ-4 adminis-tered by nursing staff; 2. Burns CP = Screened via direct interview with burns clinical psychologist; 3. MH Liaison = Screened via direct interview with the mental health liaison service; 4. Burns CP & MH Liaison = Screened via direct interviews with burns clinical psychologist and the mental health liaison service; 5. PHQ-4 & Burns CP = PHQ-4 administered by nursing staff and direct interview with burns clinical psychologist; 6. PHQ-4 & MH Liaison = PHQ-4 administered by nursing staff and direct interview with the mental health liaison service; 7. PHQ-4, Burns CP & MH Liaison = PHQ-4 administered by nursing staff, direct interviews with burns clinical psychologist and the mental health liaison service.

**Table 1.** Demographic information of those screened and those not screened.

| | Screened (*n* = 132) | Not Screened (*n* = 266) | Statistical Results |
|---|---|---|---|
| Male *n* (%) | 78 (59%) | 187 (70%) | $X^2 = 4.98$, $p = 0.03$ |
| Female *n* (%) | 54 (41%) | 79 (30%) | |
| Reviewed by hospital addictions team *n* (%) | 16 (12%) | 17 (6%) | $X^2 = 3.81$, $p = 0.051$ |
| Currently under care of a mental health team in the community (e.g., adult mental health, substance misuse, psychological services) *n* (%) | 31 (23%) | 29 (11%) | $X^2 = 10.91$, $p < 0.001$ |
| Has a diagnosed cognitive impairment (including dementia, learning disability) *n* (%) | 13 (10%) | 9 (3%) | $X^2 = 7.06$, $p = 0.01$ |
| Has a diagnosis of epilepsy *n* (%) | 9 (7%) | 17 (6%) | $X^2 = 0.03$, $p = 0.87$ |
| TBSA (median, IQR) | 4 (7.5) % | 2 (3) % | $U = 3.44$, $p < 0.01$ |
| Length of admission (median, IQR) | 12.5 (17.25) days | 6 (8) days | $U = 5.32$, $p < 0.01$ |
| Age (median, IQR) | 47.5 (26) years | 43 (28.5) years | $U = 0.67$, $p = 0.50$ |

Note: TBSA = Total Body Surface Area; IQR = Interquartile Range.

From the total number of inpatients eligible for psychosocial screening (*n* = 398), males comprised two-thirds of the sample (*n* = 265; 67%). There were 133 females recorded as inpatients (33%). When exploring gender between the samples of those screened and those who were not screened, the results found that there was a significant association between gender and psychosocial screening. This indicates that there was a lower proportion of males and a higher proportion of females screened than expected.

The association between burns inpatients reviewed by the hospital addictions team and psychosocial screening showed some evidence of an effect that did not reach statistical significance at the 5% level. There was a significantly higher proportion of screened inpatients who were recorded, either by self-report or within clinical notes, as being under

the care of a mental health team in the community at the time of admission. Similarly, there was a significantly higher proportion of screened inpatients who had a diagnosis of a cognitive impairment documented in their notes. The proportion of inpatients with a noted diagnosis of epilepsy did not differ by psychosocial screening.

Table 1 provides information about the median and interquartile ranges (IQR) for the Total Body Surface Area (TBSA), length of admission and age. This is visually represented in a box plot in Figure 2. TBSA is the percentage of body area affected by a burn injury, ranging from <1–100%. Those screened were found to have a significantly higher %TBSA than those who were not. Similarly, inpatients who were screened had a significantly longer length of admission than those who were not screened. There was no significant difference between groups in terms of age.

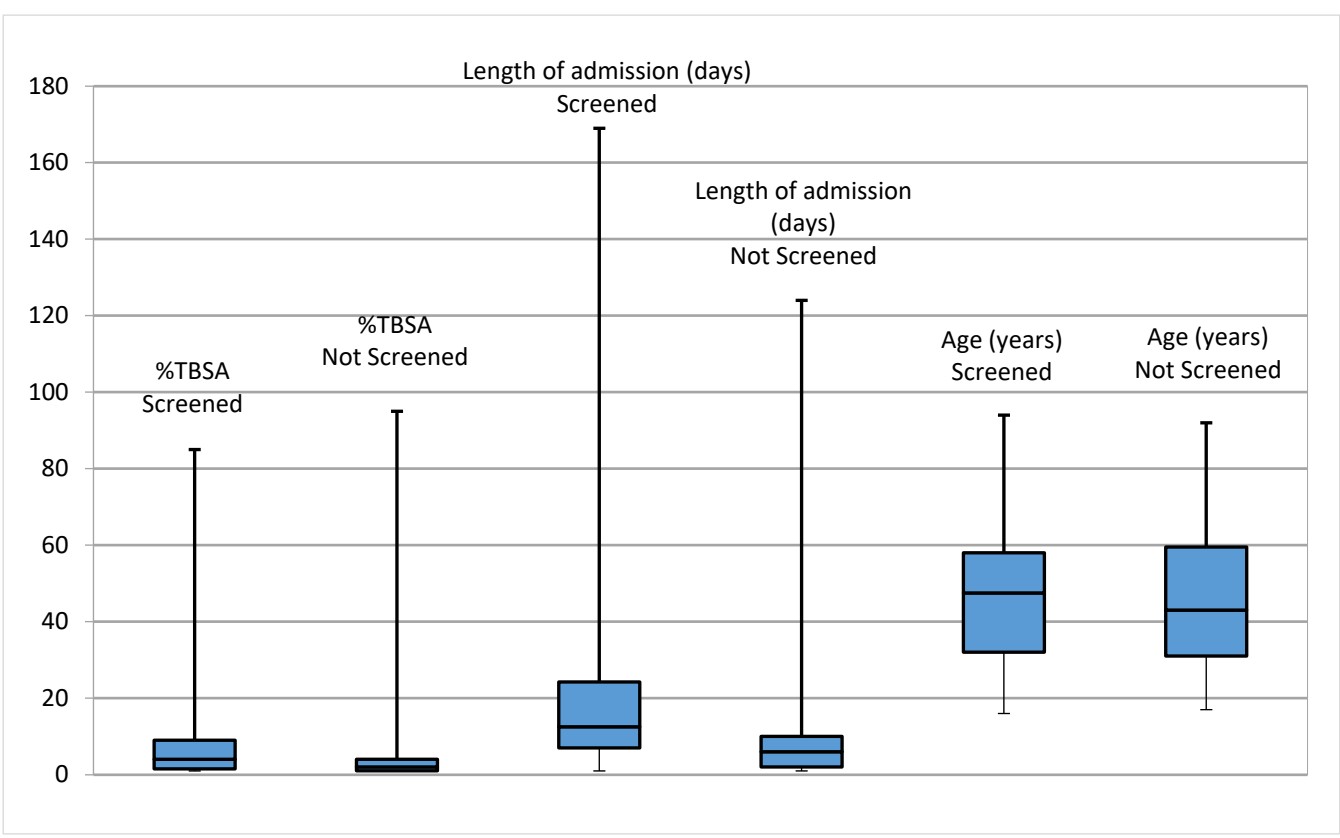

**Figure 2.** Medians and IQR of %TBSA, length of admission and age for those screened and those not screened.

## 4. Discussion

BBA Standards recommend that all patients admitted for >24 h after a burn injury be provided with a psychosocial screening assessment. Our preliminary audit data indicate that over a three-year period between November 2019 and September 2022, a third (33%) of patients were screened. The majority of inpatients were screened through the completion of a PHQ-4 questionnaire administered by nursing staff or by direct interview with the burns clinical psychologist. The audit data indicate that 67% of patients did not complete a psychosocial screening during this time period. Descriptive data were examined for the group of inpatients who were screened and those who were not in order to address the second audit question and ascertain whether there were any significant differences between the two groups.

As would perhaps be expected, those who had a longer hospital admission and greater %TBSA burns were more likely to be screened. All those who were admitted with a burns injury sustained as a consequence of self-harm were referred to the mental health liaison

service as appropriate. Inpatients with known contact with a community mental health team or a definite diagnosis of cognitive impairment were more likely to be screened. Previous research has indicated that the objective severity of injuries, such as TBSA, is not always the best indicator of psychological distress [18], and our audit indicated that those with a lower %TBSA were screened less often. More work is needed to ensure that those inpatients with smaller %TBSA burns and with no obvious mental health history or visible distress are involved in psychosocial screening.

Data show that there were significantly more females in the group who were screened. We do not know if this is because fewer males were offered or whether they were offered psychosocial screening and declined, and this needs further investigation. It was reassuring to see that there was no significant difference between the age groups of those screened and those not screened.

Having a very limited burns clinical psychology resource covering both inpatient and outpatient services means that patients can be missed if they are discharged before the psychologist is available. This highlights the importance of utilising a tiered approach within psychosocial screening to guide further assessment, psychological needs and outcomes, in the context of a limited psychology resource. As part of the National Burns Centre for Scotland implementation, an additional post (0.5 WTE) has been funded, and another clinical psychologist has been in post since August 2022. A future audit data collection will investigate the impact that this additional resource has on the psychosocial screening of inpatients.

The time period reported included the time of the COVID-19 pandemic, which put immense pressure on the staff and the service. During this period, the weekly MDT discussion was suspended, psychosocial screening via a direct interview with the burns clinical psychologist was reduced, and, for a period of one year (from the end of March 2020 to April 2021), PHQ-4 questionnaires were not administered by nursing staff. This may reflect changes in clinical priority and the possible redeployment of MDT staff during this time. All of these events understandably had an impact on screening, and the audit continues to gather retrospective data before and after the pandemic.

There are limitations in the report related to the challenges faced with data collection due to a reliance on paper notes, many handwritten, that had been scanned onto the electronic note system. There was a level of inconsistency in the detail recorded on notes regarding psychosocial screening, reflective of a busy healthcare team and individual variations in recording practices. It was not possible to include indirect MDT discussions as a method of psychosocial screening due to limited information being recorded. This may under-represent the number of inpatients screened in this way. Future recommendations include an improved recording of these indirect discussions.

We are aware that members of the MDT frequently have conversations with patients about how they are coping and about available psychological support. There was a lack of recorded data to determine whether patients had been offered screening and had declined it or whether screening had not been offered due to other factors, such as being too unwell. As a result of the audit, future recommendations include standardising the MDT data recording in order to capture whether the patient is offered a psychosocial screening and the outcome. Continued work on the audit data could include information about how long after admission psychosocial screening was completed and an exploration of the outcomes of screening within the service.

As part of being an evolving National Burns Service, future attention should be focused on the accessibility of mental health information, including those under the care of mental health teams in the community and psychiatric diagnosis. Mental health clinical notes were only available for burns inpatients residing in the locality of the burns unit, and information from those living in other localities was collected by self-report. Therefore, the numbers recorded in this report of those under the care of community mental health teams may be under-represented.

## 5. Conclusions

Our preliminary data show that during the reported time period, a third of inpatients completed a psychosocial screening, as required in order to meet the BBA Standards. Females and those already known to mental health services or with a diagnosed cognitive impairment were more likely to be screened. It is hoped that the number of screened inpatients will increase with additional clinical psychology resources. Amendments to the weekly MDT data collection sheet are planned to improve the accuracy in recording psychosocial screening information.

**Author Contributions:** Conceptualization, D.L., K.K. and R.C.; methodology, D.L. and K.K.; formal analysis, D.L.; data curation, D.L.; writing—original draft preparation, D.L. and K.K.; writing—review and editing, D.L. and K.K.; supervision, K.K. and R.C. All authors have read and agreed to the published version of the manuscript.

**Funding:** This research received no external funding.

**Institutional Review Board Statement:** Not applicable.

**Informed Consent Statement:** Patient consent was not obtained, as only anonymised and routinely collected clinical data were used.

**Data Availability Statement:** Not applicable.

**Conflicts of Interest:** The authors declare no conflict of interest.

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
