# Peer review of "Psychosocial Screening in Adult Burns Inpatients within a Scottish Burns Unit"

_2673-1991, doi:10.3390/ebj4020018_

Round 1
Reviewer 1 Report
Thank you for your work on this topic.
Introduction: Add definitions for "screening" vs "assessment." The terms are used interchangeably and need to be clarified. https://www.apaservices.org/practice/reimbursement/billing/assessment-screening
Methods: Is any screening completed for trauma-related symptoms such as acute stress d/o or PTSD?
Results:
I do not think Figure 1 is necessary. The findings are already described in text.
Was there any overlap in the methods of screening? E.g. Could a patient have been screened with the PHQ4 and by the psychologist?
Why was the median chosen as the primary statistic for the demographic and clinical characteristics? For this kind of data it is often more helpful to seen the mean and range - unless there is a cohort issue that makes the median a better choice. Please explain.
Tables: Table 1 is difficult to read in the pdf. Consider adding a column for significance values.
Lines 169-173: This is the first mention of the addiction team. The authors should consider adding more language about how patients are identified and seen by the psychologist vs the mental health liaison team vs the addiction team. This is confusing.
How long after admission was the screening typically completed?
Is any information available about psychiatric diagnoses for patients screened?
Is information available about method of injury (e.g. self-inflicted vs assault)? the authors refer to the process of caring for patients with self-inflicted injuries but do not report how many patients had this etiology.
How as it determined that patients were currently under care of a mental health team in the community? Self-report? Chart review?
Discussion:
196: Descriptive data were examined ...
204-206: This is a really important point. Glad it is highlighted.
209-214: This is a place where the authors could reiterate the importance of a tiered screening and assessment response in order to make the best use of scarce resources.
Reviewer 2 Report
This paper describes an audit of psychosocial screening for adult burn patients in Glasgow. I think this is a valuable contribution to the literature and think this paper should be accepted following some minor changes to the presentation. Well done to the authors. This paper is well written and is interesting. Thank you for the opportunity to review this paper.
Abstract
The abstract and Introduction are well written – no changes recommended. However, I do have a question that might require an addition to the text and references - Is there any information in the literature that reports the role of occupational therapists in psychosocial screening or care (lines 55 and 56)? They certainly play an important role for us in paediatric burn care.
2.3 Measures
Clarify the ways that psychosocial screening is completed by the psychologist, MH liaison (or both). If this is done via patient interview, then please add some detail – ie do they differ? one method or three?
Results:
Figure 1 is confusing, I am not sure if this refers to the team member or the method. Please clarify.
For example, if team member then:
· needs a title,
· change the label on the first bar to ‘Nursing Staff’ instead of ‘PHQ-4’
· and then relabel Figure1 as Multidisciplinary team member who completed the screen (or something similar).
Table 1
· add the comparison statistics in a column on the right of the table and remove these from the text.
· Quote actual p-values not <0.05
· The p value of 0.51 was not ‘marginal’ but “showed some evidence of an effect that did not reach statistical significance at the 5% level”. (Marginal has a specific meaning in stats speak).
Figure 2 – I don’t think this adds value and suggest removing it (or, if you keep it, add the missing tails to the box plots).
Discussion
Line 193 – was it a ‘range of screening methods’ or two (PHQ4 and interview)?
Please add a couple of sentences to list any limitations would be good to add, and then finish with a clear conclusion.
I hope these suggestions will take this to the next level in terms of clarity and presentation, and recommend for publication.
Reviewer 3 Report
Thank you for inviting me to review this short report of an audit of the numbers of patients within a burns service in Scotland who received psychosocial screening of any sort, and any differences between those who were or were not screened. Of the 398 adult patients who were eligible for screening, one third had been screened either by a clinical psychologist, mental health liaison/psychiatry, or with a standardised measure (PHQ-4). The authors found that significantly more females than males were screened, as were those with a history of being known to a mental health team or with a record of cognitive impairment. Those who were screened had a higher TBSA and a longer hospital admission. Age was not significant. The authors allude to the challenges in ensuring all patients are screened (mainly around limited resources) and the need for this to be addressed. The findings highlight the need to ensure those with lower TBSA and without a history of mental health issues or visible signs of distress are screened.
This clearly written paper could make a valuable contribution to the special edition of EBJ, and neatly complements the recently published paper in that edition by Cartwright & Pounds-Cornish (2023). However, I would encourage the authors to consider the following:
Beyond the comparison that has been made of the details of those who were screened (132) and those who weren’t (266), it would be useful to know whether there are any differences amongst those who were screened with the PHQ-4 compared with those who were screened by a mental health professional (clinical psychologist or mental health liaison/psychiatry). I would encourage the authors to consider conducting this additional analysis (it would appear they have the data for this).
They refer to the period of the audit covering the period of the COVID pandemic – some analysis of the data to consider differences in screening rates before and during the pandemic would be useful (although the authors refer to this, I found it unclear as to whether they were implying there was no screening whatsoever during the pandemic – this could be clarified).
In the PDF I received, text cuts across table 1 making the start of the paragraph illegible.
This paper reports the numbers that were screened – it would be really useful to know what the outcome of that screening was. How many were deemed to require additional support, and at what level? I imagine readers would be interested to know - is this the focus of another paper the readers could be directed to?
Round 2
Reviewer 2 Report
Thank you for your responses and for the changes to the manuscript, which I now recommend for publication. Well done.
Reviewer 3 Report
Thank you for responding to my previous comments, which have been clearly addressed in the response and revised paper. I would be pleased to see this paper accepted for publication.